# Thermal Characterization of a Gas Foil Bearing—A Novel Method of Experimental Identification of the Temperature Field Based on Integrated Thermocouples Measurements

**DOI:** 10.3390/s22155718

**Published:** 2022-07-30

**Authors:** Adam Martowicz, Paweł Zdziebko, Jakub Roemer, Grzegorz Żywica, Paweł Bagiński

**Affiliations:** 1Department of Robotics and Mechatronics, AGH University of Science and Technology, al. Mickiewicza 30, 30-059 Krakow, Poland; zdziebko@agh.edu.pl (P.Z.); jroemer@agh.edu.pl (J.R.); 2Department of Mechanical, Electronics and Chemical Engineering, Oslo Met-Oslo Metropolitan University, Postboks 4, St. Olavs Plass, 0130 Oslo, Norway; 3Department of Turbine Dynamics and Diagnostics, Institute of Fluid-Flow Machinery, Polish Academy of Sciences, Fiszera 14 Str., 80-231 Gdansk, Poland; gzywica@imp.gda.pl (G.Ż.); pbaginski@imp.gda.pl (P.B.)

**Keywords:** gas foil bearing, rotor dynamics, turbomachinery, temperature field, thermal characterization, thermocouple

## Abstract

Maintenance of adequate thermal properties is critical for correct operation of a gas foil bearing. In this work, the authors present the results of the experimentally conducted thermal characterization of a prototype installation of the bearing. A novel method of temperature identification, based on integrated thermocouples readings, has been employed to determine the thermal properties of the specialized sensing top foil mounted in the tested bearing. Two measurement campaigns have been subsequently completed, applying freely-suspended and two-node support configurations, to gather complementary knowledge regarding the bearing’s operation. Apart from the rotational speed and temperature field measurements, the authors have also studied the friction torque and the shaft’s journal trajectories based on its radial displacements. The temporal courses for the above-mentioned quantities have enabled inference on the effects present during run-up, run-out and stable state operation at a constant speed. As confirmed, the applied distribution of the integrated sensors allows for temperature readings on the entire outer surface of the foil, and therefore, provides useful data for the bearing characterization. The work is concluded with presentation of the recommended directions regarding future improvements of the proposed measurement technique and more comprehensive study of the bearing’s characteristics.

## 1. Introduction

In most of the machinery used in industry, there is a need for the installation of bearings. They are necessary to support rotating shafts and assure demanded load capacity. What is even more important, the durability of machinery depends on both the technical condition of the installed bearings and their operating parameters [1]. Amongst various types of fluid film bearings (slide bearings), the gas foil bearings (GFB)s, also known as air foil bearings (AFB)s, play an important role and provide unique and advantageous capabilities for high-speed turbomachinery [2]. Contrary to the typical slide bearings, in which the lubricating medium is primarily oil, GFBs make use of air to operate [3]. In addition, the specific properties of GFBs are even more evident when compared to the classical rolling-ball bearings [4,5]. In fact, the rolling-ball bearings are very popular because they are cheap and manufactured on a large scale in a variety of geometries and sizes. Functioning on the basis of other physical phenomena, compared to the GFBs, they withstand higher load capacities. However, rolling-ball bearings exhibit significant disadvantages related to the limited rotational speed, limited ability to transmit mechanical vibrations, tendency to increase the level of generated noise and rather quick wear [5,6]. It is worth noticing that, in contrast to the above characterized bearings, the GFBs are designed for a specific type of machinery each time from scratch, work better at higher rotational speeds and more effectively suppress mechanical vibrations—not as efficiently, however, as in case of the oil lubricated bearings. Moreover, the GFBs feature lower energy losses due to friction, tend to heat less, are quiet-running and do not wear out during stable operation.

The construction of a GFB is schematically shown in Figure 1. The most characteristic components of a GFB are the thin foils that constitute the structural part of the supporting layer for the rotating shaft, i.e., the top and bump foils [7]. While bearing’s operation, the shaft’s journal is elevated by the air film generated due to the hydrodynamic effect. Figure 2 visualizes the approximate profile of the hydrodynamic pressure which develops during run-up of a GFB. For the sake of clarity, this profile does not consider local changes in the localizations of the top foil’s support regions.

Due to the specific construction, GFBs have ability to self-adjust to the varying operating conditions [8]. It should be however noted, that GFBs are mostly dedicated to support light-loaded rotors due to the relatively high compliance of the lubricating medium used [9]. Amongst various applications of GFBs to oil-free high-speed turbomachines, in which the rotational speeds may reach 1,000,000 r/min (1 Mrpm), their use in gas and vapor microturbines and turbocompressors is well known and documented [10,11,12]. There are also known different variants of the standard construction of a GFB developed to enhance its characteristics. They make use of metal mesh, metal rubber-bump and hybrid bump-metal mesh components [13,14,15,16]. Moreover, the modified contact conditions between the bumps of the foils and the bushing, as well as the use of additional springs, are considered to change the elasto-damping properties [17,18]. Finally, various types of smart materials have been proposed to advantageously modify the thermomechanical properties of the bearings, including shape memory alloys, piezoelectric and thermoelectric materials [19,20,21]. Other structural modifications introduced to GFBs are presented in [2,22,23,24,25,26].

One of the main drawbacks of GFBs is their sensitivity to the operational temperature scatter. Specifically, a stable long-period bearing’s operation depends on the thermally induced changes of the foils’ geometry [7]. Excessive variations of the GFB’s thermal parameters may lead to the undesired mechanical deformations of the foils, and, finally, to a damage of the bearing resulting from the loss of air film due to the reduced GFB’s clearance. On the other hand, there are still gaps in comprehensive understanding of various effects observed during operation of a GFB which are activated by changing temperature. The thermally induced changes of the resultant elasto-damping properties, load capacity and friction coefficients have been confirmed in [8,27,28,29]. The relationships between the rotational speed, the assembly preload and the temperature are discussed in [30,31]. Moreover, an overview on the modeling methods for GFB, including temperature simulations, can be found in [2,9,32].

The high significance of the thermally induced phenomena in GFBs became the motivating factor for the authors of the present paper to conduct high spatial resolution temperature measurements with the use of specialized sensing top foil. The recently applied methods for the temperature characterization mostly make use of infrared cameras and industrial thermocouples, preferably mounted at several points of the bearing’s structure only [9,33], and also in the case of thrust GFBs, as presented in [34]. However, more reliable installation techniques, also allowing for denser sensor distribution, are required. The main objective of the current study is to experimentally validate the proposed novel measurement approach which is based on the readings provided by the thermocouples integrated with the top foil [35]. In the authors’ opinion, the temperature measurements performed with the tested prototype installation will allow for investigating the effects present in GFBs more deeply, e.g., identifying the multiphysics relationships and, finally, aid in determining the conditions necessary to maintain a stable operation of the bearing.

The paper is composed as follows. After the introductory Section 1, a novel method and the constructed GFB prototype installation applied for thermal characterization are described in detail in Section 2. Section 3 introduces the experimental part of the conducted research including a description of the two measurement configurations. Next, Section 4 and Section 5 present the results of the performed experiments, followed by their discussion in Section 6. Section 7 summarizes the work and draws the conclusions.

## 2. Novel Method for Experimental Temperature Field Identification in GFB

In the following, a novel method for the experimental identification of temperature field in a GFB as well as the prototype of a specialized sensing top foil equipped with the integrated (built-in) thermocouples are presented, making reference to the authors’ previous published works. As mentioned in Section 1, the research field of the GFB’s thermal characterization still exhibits lack of technical solutions that would provide both reliable and high spatial resolution temperature measurements. Industrial thermocouples, even if theoretically allowed to be densely installed on the bearing’s foils to assure high resolution of the temperature readings, would introduce an unacceptable risk of their detachment during the GFB’s operation. To the authors’ best knowledge, there is no known application reported in the literature, in which the number of installed temperature sensors of any type is equal or greater than 18, as considered in the GFB’s prototype installation proposed by the authors of the present work.

The actual novelty of the presently investigated approach is that a special type of the top foil, which is equipped with integrated thermocouples, is used for temperature measurements in a newly designed GFB that has a unique construction. This is a new approach to the temperature identification in GFBs, not found in other applications of this type of bearings. Effectively, the top foil becomes a sensing structural component capable of reliably registering the temperature on its entire surface. The proposed method of measurements advantageously eliminates the classical problem of continuous maintenance of the mechanical contact between standard, preferably industrial, thermocouples and the inspected object. Specifically, for a standard case of typical thermocouples, it is difficult to keep constant and low thermal resistance between the sensor and the object being tested that would assure long-term and valid data acquisition.

The concept of a specialized top foil was first introduced in [35]. The technological details on manufacturing the integrated thermocouples are addressed in the above referenced work. The test stand presented there, however, suffered from disadvantageous wiring distribution and did not consider appropriate bearing’s structural adaptation that would allow for its reliable long-term operation. In fact, one of the main drawbacks of the referenced prototype was the longitudinal orientation of the cables connected to the thermocouples integrated with the top foil. Specifically, these cables were coming out of the GFB through the spaces between the bumps of the bump foils which would not be acceptable in an industrial application. In contrast, the radial guidance of thermocouples wiring, as well as the unique construction of a recently designed GFB prototype, enable convenient and reliable temperature identification [36]. The newly-designed bearing has a three-part bushing, as presented in Figure 3. Its nominal diameter equals 30 mm.

It is important to note that before a specialized sensing top foil and the adapted GFB were manufactured, the authors conducted preliminary FE simulations making use of the developed CAD model of the three-part bushing, top and bump foils. The results of numerical simulations reported in the work [37] allowed us to initially assess the temperature distribution within the bearing’s bushing. This study enabled to find the extremes of the mentioned quantity and, hence, to aid the design process for the specialized top foil, specifically in terms of rational selection of spatial distribution of the integrated temperature sensors. The temperature field was simulated for the GFB’s nominal operational conditions. The details regarding numerical calculations and model properties, specifically including the definition of boundary conditions, can be found in the above-cited paper.

The results obtained during the referenced FE analysis have motivated the authors to uniformly cover the entire outer surface of the top foil with integrated thermocouples. Effectively, three sets of six thermocouples each, i.e., 18 sensors in total, have been proposed along the circumference of the bearing, as presented in Figure 4. The prototype of the specialized sensing top foil is shown in Figure 5. At this point, it is also worth mentioning that apart from the thermocouples, strain gauges have also been mounted in the prototype, as presented in Figure 5. Nevertheless, strain measurements are considered as out of the scope of the current work.

Both the top and bump foils are made of the superalloy Inconel 625 that exhibits advantageous mechanical properties. It can be significantly elastically deformed which is demanded in a GFB. Moreover, it is a highly corrosion resistant material. The unique construction of the sensing top foil has required designing an innovative prototype of the GFB, i.e., making an adequate adaptation of the classical bearing’s construction. The parts of the newly designed GFB’s prototype are shown in Figure 6.

The specialized sensing top foil and three bump foils have been installed in the bushing. The bushing consists of the three tierces to allow for a convenient installation of the bearing’s foils equipped with radially oriented wiring connected to the integrated thermocouples. The sensors’ cables have been routed out of the GFB though 18 openings made in the bushing’s tierces. Finally, the fully assembled bearing’s bushing has been prepared for the installation in the test stand, following the measurements’ configurations presented in Section 3.

## 3. Experiments–Investigated Cases

The experimental part of the study has been performed for the investigated GFB using the two standard measurement configurations [32,38]. Specifically, while initiating experiments for a newly-designed and constructed bearing, a common testing procedure is introduced, considering: (1) a freely-suspended (also known as floating) configuration, and (2) two-node support configuration. Figure 7 and Figure 8 present schematic views of the two above-mentioned measurement configurations.

These configurations provide complementary knowledge regarding the operation and various properties of the tested GFB, gathered sequentially during the prototyping procedure. What is important to be mentioned here is that the first configuration (a freely-suspended configuration) assumes assembly of the tested GFB without any housing that would force its rigid fixation. In the other words, the bearing is allowed to adjust its position and orientation freely, being only captured with an auxiliary handle. Contrarily, two supports are considered in the case of the second configuration (two-node support configuration), where one of the two assembled bearings becomes the tested one. Moreover, these two configurations have opposite angular orientations of the installed foils. In the former measurement case, the bushing is supported by the shaft. It means that the bottom localization of the top foil tie is considered, i.e., under the shaft, that assures development of the hydrodynamic pressure on the upper part of the bearing. By doing so, during run-up the bushing is allowed to start elevating over the shaft. An inverse localization of the top foil tie is taken into account for the later configuration as the shaft is supported by the two bearings, in turn.

The first configuration assures a safe course of the running-in process conducted for a new top foil operating for the first time. The auxiliary handle stands for a compliant fixation and, hence, allows more freedom in the foil’s geometric adjustment. The freely-suspended configuration enables identification of the friction torque with the force sensor. Moreover, the rotational speed as well as the temperature distribution in the top foil can be determined in the test stand.

The second configuration introduces stiff fixation for both bearings. It is considered to reliably represent operational conditions found in real applications. The installed measurement gauges allow determination of the shaft’s trajectories making use of the radial displacements identified by the transversely oriented 1-D proximity sensors. Apart from the above-mentioned quantities, again, the rotational speed and the temperature field can be read.

The measurements conducted for the above discussed two configurations are addressed in the following, in Section 4 and Section 5, respectively. In both cases the software National Instruments LabVIEW was used for data acquisition and analysis performed for the temperature and speed of rotation measurements. The temperatures were registered with the hardware module National Instruments PXi-NI4353 whereas the device LMS SCADAS Mobile 05 was employed for trajectory and friction torque readings acquisition. The mentioned LMS hardware was controlled with the software LMS Test.Lab that stored and visualized the acquired data.

## 4. Measurement Campaign I

The first measurement campaign has been conducted for the freely-suspended GFB. Hereunder, Section 4.1 and Section 4.2 address both the laboratory test stand used and the obtained results.

### 4.1. Laboratory Test Stand

The laboratory test stand is presented in Figure 9 and Figure 10. It is composed of a high-speed motor (electrospindle) with its inverter, the tested GFB and the shaft. The accompanying measurement instrumentation allowed identification of the friction torque, rotational speed and temperature for the top foil.

As shown in Figure 9, the shaft is directly connected with the motor without a clutch, as this component is not required in the case of the auxiliary fixation applied to the GFB’s bushing. The friction torque was found via readings from the force sensor mounted on the arm that holds the bearing. The sensor is localized at the radius of 125 mm with respect to the shaft’s axis. The rotational speed was acquired with an optical sensor.

Various testing conditions have been applied during experiments. In the following the outcomes obtained for the first run after installation of the GFB are presented, assuming trapezoidal temporal course for the rotational speed of the shaft. It has been assumed that the run-up stage of the operation lasts for 30 s until the electrospindle reaches the nominal speed of 24,000 r/min (24 krpm). Hence, the speed rate equals 800 r/min/s (800 rpm/s). Next, operation at the constant speed is performed for approximately 560 s. Finally, the run-out stage is carried out within the period of 6 s—with the speed rate—−4000 r/min/s (−4 krpm/s). The authors admit that the speed of 24,000 r/min (24 krpm) is rather low for the considered type of bearing; however, as has been experimentally confirmed, the achieved operational conditions are appropriate to efficiently generate the air film and study the characteristics of the tested prototype installation.

### 4.2. Results

Figure 11 presents the temporal courses for the measured quantities, i.e., rotational speed, temperatures (including the minimal, maximal and mean values) and friction torque. Moreover, for the selected time moments the temperature profiles are visualized in Figure 12. For better visualization of the results, the top foil is shown flattened. The presented temperature fields are linearly extrapolated and interpolated based on the thermocouples’ readings. The corner of the top foil where the thermocouples’ common electrode and the free end of the shaft are localized is also marked in all plots in Figure 12, using the label “TC electrode”.

As previously mentioned, the authors used interpolation and extrapolation approaches to reconstruct the temperature values at both the interior and edge points of the top foil. In fact, in the currently investigated test stand, it is not feasible to mount the sensors directly at the top foil’s edge. Nevertheless, the temperature estimates are provided for the entire surface of the bearing’s tested component for a general reference, making use of straightforward linear approximation functions. However, to provide a more reliable experimental inference on the GFB’s properties, the authors consider during the future planned studies to complement the presently available thermocouple readings with the outcomes of the measurements at the foil’s edge points conducted with an infrared camera.

The first operation of the GFB prototype, i.e., performed right after the bearing’s assembly has been completed, is interesting since it provides data regarding the processes of top foil running-in and the adjustment between the bearing’s foils and the shaft. The measurements show an uneven temperature profile for the top foil which corresponds to the investigations reported in the literature [39], specifically with respect to the temperature change observed along the bearing’s axial direction. A detailed discussion on the measurement results can be found in Section 6.

## 5. Measurement Campaign II

Similarly, the second measurement campaign is addressed hereunder, having introduced the description of the laboratory test stand and the registered results in Section 5.1 and Section 5.2, respectively.

### 5.1. Laboratory Test Stand

Figure 13 and Figure 14 present the test stand and the measurement instruments used for the two-node support bearing configuration. Apart from the components considered in the previous case (described in Section 4), i.e., electrospindle, inverter and the tested GFB with a shaft, there are also installed an additional bearing (marked as ‘Bearing 2’ in Figure 13) and a clutch.

The measurement instrumentation enables acquisition shaft trajectory via its two transversal radial displacements, rotational speed and temperature for the top foil. The shaft’s radial displacements are measured making use of the two pairs of laser and inductive proximity sensors mounted at each of the two bearings.

In the considered measurements, the clutch is mounted to suppress axes misalignments between the motor and the shaft supported by the two GFBs. It should be however noted that, prior to the connection of the shaft with the motor via the clutch, the two bearings were positioned so that their mutual misalignments did not exceed the allowed level assuring the correct operation of the test stand. Improper installation of the bearings and motor would inevitably lead to the top foil damage due to tearing off the protective polymer layer. The considered measurement configuration represents a realistic case of industrial application where at least two bearing nodes are taken into account and, hence, the top foil ties are localized on top of the bushings as visualized in Figure 8.

Amongst various testing conditions considered during measurements, the results obtained for an asymmetric trapezoidal temporal course for the rotational speed of the shaft are presented in the following. The assumptions about the run are as follows: the period of a low-speed rate run-up stage at 80 r/min/s (80 rpm/s) equals 300 s, the nominal rotational speed at stable operation is 24,000 r/min (24 krpm), the period of stable operation at constant speed is 300 s, and the period of a run-out stage equals approximately 50 s and has been performed at the average speed rate −480 r/min/s (−480 rpm/s). There was no external load attached to the shaft in the considered case study.

### 5.2. Results

Figure 15 presents the temporal courses for the measured quantities, i.e., rotational speed, the three characteristic temperatures and the shaft’s radial displacements. Again, for the selected time moments the temperature profiles are visualized in Figure 16. Figure 17 presents the shaft’s trajectories registered during stable operation, i.e., within the time interval 300–600 s.

It should be noted that the tests for the investigated GFB have been performed using the run-in surface of the top foil as well as the final geometries of all bump and top foils. It means that after a sufficiently long period of operation, these components already corrected its shape and surface to the geometry and position of the shaft. The identified scatters for the X and Y displacements determined based on the measured signals equal (0.744, 0.223) and (0.292, 0.258) micrometers, respectively, for the two mounted bearings. The scatters counterparts of the X- and Y-axis displacement found for the filtered data take the following values: (0.406, 0.125) and (0.117, 0.151) micrometers. A detailed analysis regarding the registered results can be found in the following Section 6.

## 6. Discussion

During the first measurement campaign the specialized top foil had not been yet run-in. This state of the foil significantly influenced the behavior of the tested GFB. Specifically, high temperatures and their scatters were registered during the entire experiment, with the maximum values as high as approximately 100 °C.

The authors observed characteristic growth of the torque for the dry friction stages of operation (Figure 11). First, while gradual speeding up (run-up) until the air film is completely developed, the bearing experiences high friction due to a direct contact between the shaft’s journal and the inner surface of the top foil. Next, the friction rapidly drops and maintains at a low level for the nominal rotational speed. Moreover, during the mentioned stage of stable operation, as an effect of the ongoing process of running-in of the top foil, the registered force successfully reached even lower values. Exceptionally, approximately at the experimental time equal to 550 s, short-period peaks appeared during the discussed stage of operation. In the authors’ opinion, they occurred most likely because of the progressive process of gradual geometric adjustment of the GFB’s foils. Finally, raised friction was also found during run-out, again passing through the specific speeds of rotation after the air film was lost. The above-formulated observation regarding the torque also finds confirmation in the temperature course. Similarly, for the second half of the stage of stable operation, a gradual decrease of the temperature at the constant rotational speed was also experienced, which also confirmed the ongoing desired geometric shape adaptation of the bearing’s foils. Figure 18 presents complementary results for the run-in bearing’s top foil. Constant values of the measured temperatures confirm the completed process of the shaft’s journal adjustment.

It is important to mention that in Figure 11 a short-period loss of correct signal from one of the thermocouples occurred. Since an initial run for a given top foil can be performed only once, this artifact, i.e., an outlier, was allowed to be presented in the reported results. It should be explained here that no additional experiment would enable repeating the initial running-in procedure for a given set of the newly-installed bearing’s foils. The process of geometric adjustment that takes place in a GFB is irreversible, specifically in terms of the accompanying running-in observed within the protective surface in a top foil.

The significance of the course of the running-in process is also manifested via considerably high axial scatter of the temperature measured within the top foil, as seen in Figure 12. The temperatures identified at the foil’s edge localized close to the motor, are characterized by higher values than the ones present at the free end on the shaft, i.e., where the thermocouples’ electrode can be found. The authors also initially conclude that the heat generation that originated from a dry friction present during adaptation of the foil’s edge, within the above-mentioned process of running-in, effectively led to asymmetry of the temperature field. Additionally, this undesired effect was gained by the flow of the heated air coming out of the electrospindle due to its forced circulation. In fact, more insight can be now provided on the specificity of the initial stage of the foil’s operation. The authors, however, are aware of the need of further investigation of the raised issues to be able to formulate more unambiguous conclusions about the running-in process kinetics and consequences.

Proper axial alignment between the shaft and bearings as well as use of a run-in top foil is crucial to maintain a long-period reliable operation of a GFB. The above-mentioned issues considerably influence the temporal courses for the temperatures measured within the top foil. In fact, geometric mismatch of the shapes of cooperating components results in a dry friction and sudden temperature growth. This phenomenon is clearly seen for the first run of the top foil, that was registered for the freely-suspended bearing’s configuration and visualized in Figure 11 and Figure 12. It should be noted, however, that further wear of the top foil and continued loss of the protective polymer cover depends more on the number of run-up/run-out cycles than on the length of the operation period while stable operational conditions are held.

Considering the stage of stable operation during the second measurement configuration, the run-in top foil assures significant decrease of both mean values of the temperatures (approximately from 90 °C down to 40 °C) and their scatters (from 20 °C down to 10 °C) as shown in Figure 15 and Figure 16. It is also seen that in the case of the second measurement campaign the temperature advantageously decreases after the nominal speed is achieved, contrarily to the first measurement case. More uniform spatial distribution of the temperature is found as well.

There are also observed stable in time and low-valued scatters of the shaft’s X and Y radial displacements at the inspected bearing during the stage of nominal rotational speed (Figure 15 and Figure 17). The registered values of these scatters are of the same order as the ones obtained for a standard construction of a GFB. It should be noted that the value found for the X-axis readings remains three times as high as the rest of the indicators; however, all these quantities are found within the limit of 1 micrometer, which is acceptable taking into account the thickness of the generated air film, i.e., at least several micrometers. Moreover, the authors think that the asymmetry of the trajectory plot that showed up for the investigated GFB results from the initial deformation of the top foil happened either during bearing’s assembly or sensor installation. It turned out that the nonstandard treatment of the foil, to enable it to become a sensing component, visibly affected the operation of the bearing. In fact, local changes in damping and stiffness properties were probably induced during the construction of the prototype. However, these factors do not prevent the GFB from operating normally. What is even more important, the results confirm the validity of the proposed concept of a specialized sensing top foil installation, even though a relatively large area of the top foil remains unsupported, i.e., without bump components underneath. As experimentally proven, the interference in the bearing’s structure was not large enough to make it stop working.

The authors would like to share some of the important aspects related to the inconveniences and difficulties experienced during measurements and prototype assembly. Undoubtedly, the high number of integrated thermocouples opens an opportunity to analyze the temperature distribution within the top foil more comprehensively than using only few industrial sensors. However, a problem with guidance of all the sensor cables arises. In fact, there is not much room in the GFB to be occupied by wiring. Therefore, the possibility of using many sensors in the prototype installation has forced significant changes in the design of the bearing.

Moreover, even though the thermocouples assure effective and reliable thermal contact at low thermal resistance, they are fragile and may be easily damaged (broken off), especially in presence of many cables touching one another. Nevertheless, the sensors can be repaired despite difficult access. The described problem appeared during both the construction of the test stand and its operation, due to vibrations. The assembly process of a GFB equipped with a specialized top foil is therefore time-consuming. The authors are aware of the need for future reduction of the number of sensors and improved wiring fixation to allow for the preparation of an industrial application. This will be possible, however, after the study on the relationship between the selection of the location of temperature measurement points and the significance of the obtained results in terms of the bearing’s condition assessment is completed. It is also worth noting that a calibration procedure for the thermocouples may be an issue since the technique used for their installation does not guarantee the repeatability of their parameters.

Finally, in case of the freely-suspended configuration it is recommended to mount a cover that will protect the inspected bearing from the air heated by a motor. Reduction of the thermal effects originating from an additional hot air circulation should lead to more effective control over the GFB’s operational properties.

The authors admit that the resultant temperature distribution and, therefore, the bearing’s operational conditions, may change due to the fluctuations of the speed, cooling air flow rate as well as the loading, also considering the influence of the possibly exiting interactions between these factors. More comprehensive study on this topic is scheduled by the authors for future research.

The temperature sensors may be spread out only within the areas where the three bump foils do not touch the top foil. Otherwise, the integrated thermocouples would surely disturb a proper interaction between these foils, i.e., they would disadvantageously interfere with the contact conditions and prevent the necessary micro-sliding of the foils. The geometric limitations resulting from the above-mentioned operational issues makes it impossible to mount the temperature sensors at any location on the outer surface of the top foil. On the other hand, any demand for arbitrary localizations of the thermocouples requires adequate local modifications of the bearing’s structural part of the supporting layer to allow for sensor installation. Consequently, a new geometry for the bump foils is required to assure enough space for the welded thermocouples as was considered in the proposed measurement technique. In effect, the prototype GFB’s elasto-damping properties were locally changed. Therefore, the profile of bending deformations for the top foil is modified during the bearing’s operation due to the introduced geometric revision, which is then followed by the respective slight changes of both the air film height and the temperature in the considered regions. In reference, one of the main goals of the present study was to confirm the validity of the proposed measurement approach that inevitably leads to the above-mentioned structural modifications. However, this approach was successfully tested since the investigated prototype bearing’s installation worked properly. The air film was developed and, next, it elevated the shaft’s journal as was confirmed with the registered data, i.e., via both the temperature and friction torque drops which are seen in Figure 15. Moreover, the authors are aware of the fact that the temperatures identified for the entire top foil, via interpolation procedure, tend to be underestimated due to the above-discussed phenomena present at the localizations of measurement points, resulting from the applied structural modifications. Concluding, the proposed measurement technique stands for a compromise between the structural requirements, understood as the allowed modifications of the bump foils’ geometry that still assures proper operation of the bearing, and the desired spatial distribution density for the integrated temperature sensors. It should be mentioned that, selective, i.e., local and periodical support for the top foil which is assured by the bumps of the bump foil is a general property of GFBs. It is natural that the entire surface of the top foil is not supported with bumps, even when no structural modification of the bearing is taken into account compared to the typical, standard construction of a GFB. Addition of thermocouples, i.e., the caused size reduction of the bump foils enhances the effect of local change of the elasto-damping characteristics. Moreover, additional unsupported regions may be intentionally introduced between the bumps so that the bump foils’ stiffness drops and to allow these foils to operate independently. Finally, the discussed structural change, applied within reasonable limits, is not a factor that would cause modification of nature of the bearing’s work.

In addition, the rotating shaft itself may significantly affect operation of the bearing, as discussed in [40], where the results of the studies conducted for the class of ball bearing systems are provided as reference. The above-mentioned influence originates from the shaft’s flexibility. Its undesired deformations, resulting from the excited normal modes, preferably the bending ones, may disadvantageously interfere with the contact characteristics of the bearing, and finally, with its proper operation. However, as investigated by the authors of the present paper, the raised issue does not apply to the analyzed experimental work. In fact, even for the longer and, therefore, more flexible shaft, supported with the two bearings during the second measurement campaign, the first critical speed of rotation, identified via numerical calculations, was about 45,000 r/min (45 krpm), which is much more than the maximum speed of the electrospindle considered for the experiments, i.e., 24,000 r/min (24 krpm). Therefore, it is concluded that the two shafts used in both measurement campaigns may be considered as rigid and, hence, no bending vibrations are expected to be observed in the investigated test stands. In consequence, the authors assumed that for the entire tested speed range there were no such large bending deformations of the rotating shaft that could cause uneven distribution of loads and deflections of the bearings’ foils and finally cause significant misalignment of the shaft’s journal with respect to the bearing’s bushing. In the presented research, the effect of the rotating shaft is considered as negligible.

There are couple of practical issues related to the measurement errors that are worth discussion. First of all, since the proposed approach is based on the concept of integrated sensors, it assures a common electrode for all installed thermocouples, i.e., the same referential temperature may advantageously be considered for all measurement points. However, the procedure of automatic compensation dedicated for standard industrial thermocouples, which is available in the utilized 24-bit temperature measurement module National Instruments PXi-NI4353 within the built-in electronic circuit equipped with a thermistor, may not be applied. Hence, the custom settings for the welded sensors were considered by the authors, following the readings from the calibration procedure. Basically, the measurements of voltage, specifically thermoelectric forces, were performed for all integrated thermocouples and then converted to temperature readings. Consequently, no access to the thermistor measurements was provided for the used mode of operation of the National Instruments module, unfortunately. Therefore, a constant value of 25 °C was declared as the cold-junction temperature. Nevertheless, the changes of the ambient temperature, that become the same for all sensors, registered during the experiments were negligible, as caught within the range ±1 °C. Effectively, neither the absolute nor relative values of the measured temperatures were significantly affected by the characteristics of the employed measurement technique. It should be mentioned that even though the custom-made thermocouples themselves provide accurate temperature readings via generated linearly-scaled voltages, there are measurement errors that depend on the quality (accuracy) of the performed calibration procedure and quality (resolution) of the used analog-to-digital data processing. The calibration procedure was performed for a series of temperature measurements and then the linearly interpolated and extrapolated temperature readings were calculated for actual values of the registered thermoelectric forces. For both sources of the measurement errors, i.e., the calibration procedure and data acquisition in the National Instruments module, the error is estimated by the authors as not exceeding the range ±0.5 °C for the temperature readings from 0 °C up to 100 °C. Concluding, the authors would like to mention that in the case of the proposed measurement technique a more detailed and reliable inference on the propagation of measurement errors and their budget estimation, is considered for the future study after presentation of the results of all preliminary experiments conducted for the GFB’s prototype installation.

## 7. Summary and Final Conclusions

Being motivated by the discussed drawbacks of the currently used measurement methods and technical solutions dedicated for the characterization of temperature distribution in a GFB, the authors of the present work designed and constructed a novel prototype bearing’s installation equipped with the specialized sensing top foil hosting integrated thermocouples.

The developed test stands successfully underwent the two measurement campaigns making use of both the freely-suspended and two-node support configurations. These two complementary experimental studies have allowed us to draw conclusions about various properties and capabilities of the newly-designed bearing. The conducted research has enabled us to register the temporal courses of the rotational speed, torque friction, temperature and the shaft’s trajectories with radial displacements.

One of the main achievements of the study is the confirmation of a correct operation of the prototype GFB installation making use of the two above-mentioned measurement configurations. The paper presents the proof of concept for the method of temperature identification based on the measurements in a unique prototype bearing system. The applied distribution of the integrated sensors allows for temperature readings on the entire outer surface of the foil.

In the authors’ opinion, it is also worthwhile to summarize the novel aspects of the conducted research. As mentioned in Section 2, to date there is no known other study reported in the literature that would provide such a specific means for experimental temperature investigations for an operating GFB as presented in the current work. The following unique properties and capabilities are provided thanks to the proposed and successfully applied technical solution:Application of a foil-sensor, i.e., the use of a specialized top foil with integrated thermocouples providing the capability of a direct use of the material of the foil–acting as one of the two required components (electrodes) of a thermocouple. In the other words, the top foil itself becomes a temperature field sensor. A new temperature measurement method and physical setup for more reliable temperature field identification has been successfully experimentally tested.Temperature measurement performed directly on the top foil surface, specifically on its outer surface, since this component of the structural part of the bearing’s supporting layer is a sensing component, as mentioned above. This approach allows for more accurate temperature measurement, i.e., effective at a smaller distance with respect to the region where the air film develops and operates elevating the shaft’s journal. The foil is thin–its thickness equals 0.1 mm. Hence, it is expected that the temperature distribution estimated across its thickness is uniform. There is no additional material used to build a sensor apart from the platinum alloy, becoming one of the two electrodes in a given thermocouple, after being welded to the foil. In the other words, no additional material is considered for the phenomenon of heat energy flow during the physical process of temperature measurement.Capability of dense measurement points to localizations within the top foil that provides more accurate identification of the temperature spatial distribution.High and advantageous integration of the sensing component with the inspected device which brings closer a future perspective of the development of structural health monitoring based solutions for GFBs and building of temperature control-oriented systems.Reliable installation of the temperature sensors assuring their firm fixation within the thermal contact areas and, therefore, less sensitivity to mechanical vibrations of the shaft. Small mass of the electrodes made of platinum wires, of diameter 0.1 mm, results in low mechanical strength originating from the generated inertial forces. The sensing nodes made with industrial thermocouples are more prone to shifts and damages.Less sensitivity of the measurement results to temperature sensor installation errors considering as reference the mounting techniques available for industrial thermocouples. Neither the use of adhesives nor mechanical holding applied in case of the above-mentioned sensors provides as reliable and accurate measurements as the ones available in the proposed solution. In case of industrial sensors there is a serious risk of continuous change of the contact conditions due to either the presence of an additional material (an adhesive) or fluctuation of the holding force. Consequently, despite potentially high accuracy and repeatability characterizing the measurement conditions for the industrial thermocouples, the quality of the conducted temperature measurement may be significantly affected by the bearing’s behavior during its operation and, therefore, questionable. The above conclusions are formulated based on the authors’ previous experiences gained during installation of industrial thermocouples in the investigated GFBs’ prototypes. In contrast, the developed sensing foil based approach leads to convenient research inference since the welded thermocouples tend to either work properly indicating the correct values of the temperatures or do not provide any realistic value at all while being damaged. This nature of a sort of binary operation assures immediate detection of sensor malfunction. Industrial sensors may gradually lose mechanical and, therefore, thermal connections. Finally, it may be difficult to estimate the measurement error when gradually losing a sensor’s connection with the top foil. In the authors’ opinion, the described property is one of the most important for the proposed measured technique.Measurements of temperature fields are conducted for a normally operating bearing (specifically in the case of the second measurement campaign) and allows for tracking the changes of measured quantities during both development of the air film and its loss. It is important to mention that, apart from the design proposal and prototype development, new and unique experimental results have been obtained and presented in the current work for the selected operational conditions of the tested bearing which may be used as reference for both verification and validation of the numerical models. The current work makes a reference, experimental verification, continuation and significant extension with respect to the authors’ previous studies regarding both: (a) experimental proof of concept for the integrated temperature sensors presented in [35] (noticing that the preliminary results reported in the cited work were acquired for a non-rotating bearing, i.e., the GFB that did not normally operate, and the applied longitudinal guidance of the sensor wiring made it impossible to use the developed bearing’s prototype installation in a practical industrial solution), and (b) the concept as well as the CAD model for the newly-designed bearing composed with a three-part bushing allowing for a practical use of the above-mentioned temperature measurement method [36].The presented unique technical solution is dedicated for further development towards industrial applications. However, additional issues should be addressed, e.g., choice of the sufficient number and localizations of the integrated thermocouples as well as their protection against operational conditions.

Moreover, there are several aspects worth addressing, confirming the usefulness of the obtained experimental results:
Spatially dense measurements for the temperature fields Performed accurately and directly on the top foil allow for a more realistic, as experimentally proven, mapping of the mentioned quantity dedicated for developing new numerical models of GFBs and to provide a new tool to investigate the thermal behavior of the bearings, e.g., for tracking the processes of generation and losing the air film, especially when no data on the friction torque is available for reference.The conducted measurements for the temperature fields allow for more reliable inference on the operational conditions of the tested bearing, e.g., in terms of identification of critical temperature gradients leading to GFB damage, and enable tracking these characteristics during the bearing’s operation with high spatial density.The accurately experimentally identified localizations of the temperature extremes, i.e., the minimum and maximum values, may allow for future guidance formulation regarding GFB’s construction modifications, including suggestions on local changes of the bump foil’s stiffness aimed at improving the temperature distribution. Having introduced a more uniform temperature field, an even mechanical load should also be available that may finally lead to reduction of the bearing’s wear and acceleration of the process of air film development.The results of the performed measurements may be used for developing a system that controls the state of the bearing’s operation and detects its anomalies.

Finally, it should be highlighted that the presented results were obtained for various operation regimes and bearing configurations, i.e., the freely-suspended bearing and the two-node support configuration, and are considered by the authors as examples presenting the achieved capabilities of the investigated prototype test stand and the configured measurement equipment.

As regards future investigations, the authors will consider the assessment of the temperature measurement repeatability and possibly uncertainty of propagation quantification carried out for varying operational conditions as well as study of the influence of geometric and material imperfections on the bearing behavior, specifically in terms of working stability. Moreover, an experimental validation for the elaborated FE model is scheduled to be performed to propose new usable guidance for GFB model development. Finally, the authors are planning to investigate the thermomechanical coupling present in the tested GFB, making use of the strain gauges also mounted on the top foil.

## Figures and Tables

**Figure 1 sensors-22-05718-f001:**
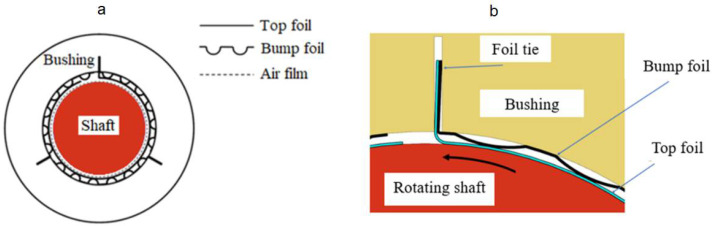
Construction of GFB: (**a**) cross-sectional view, (**b**) magnified view of the top foil tie region.

**Figure 2 sensors-22-05718-f002:**
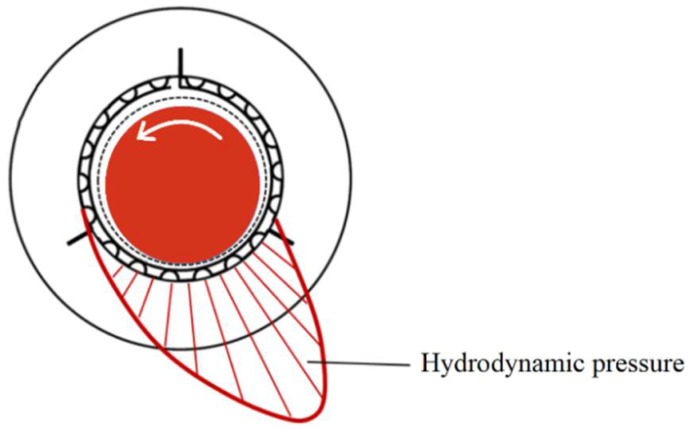
Approximated representation of the region where the hydrodynamic pressure is developed due to the shaft’s journal rotation in GFB.

**Figure 3 sensors-22-05718-f003:**
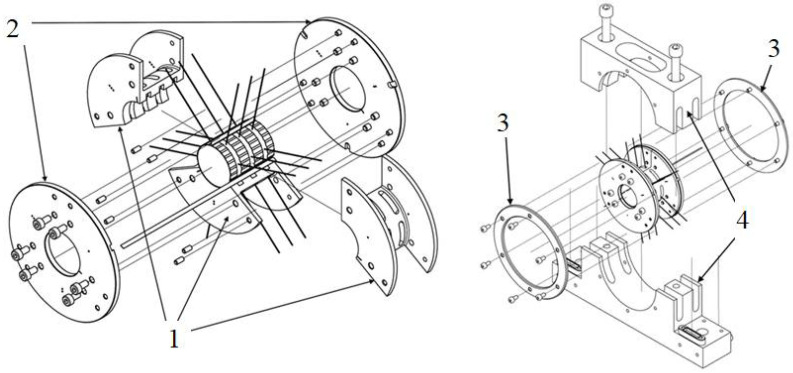
CAD model of the GFB dedicated for the application of the specialized sensing top foil (adapted under a Creative Commons Attribution 4.0 International License-https://creativecommons.org/licenses/by/4.0/ [36]). The presented components are: 1-tricuspid bushing shell, 2-thrust flange with screws, 3–thrust rings, 4-bearing housing. The wiring marks localizations of the integrated thermocouples.

**Figure 4 sensors-22-05718-f004:**
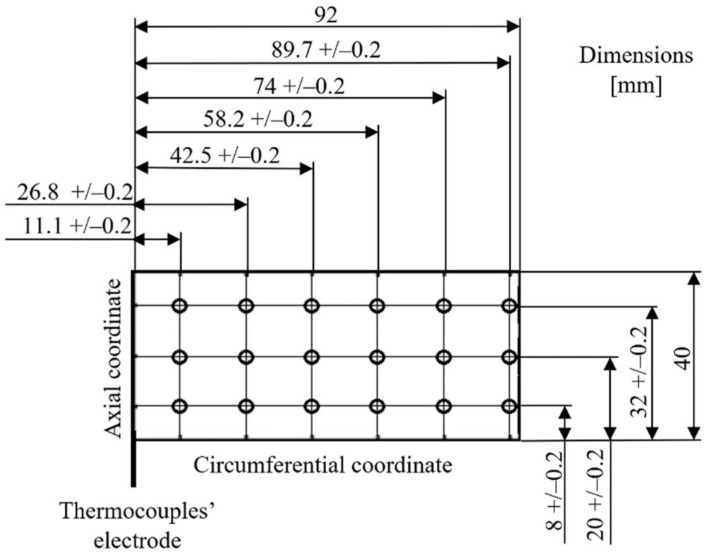
Spatial distribution of the integrated thermocouples within the top foil. A planar view for the flattened foil is used to clearly mark circumferential dimensions.

**Figure 5 sensors-22-05718-f005:**
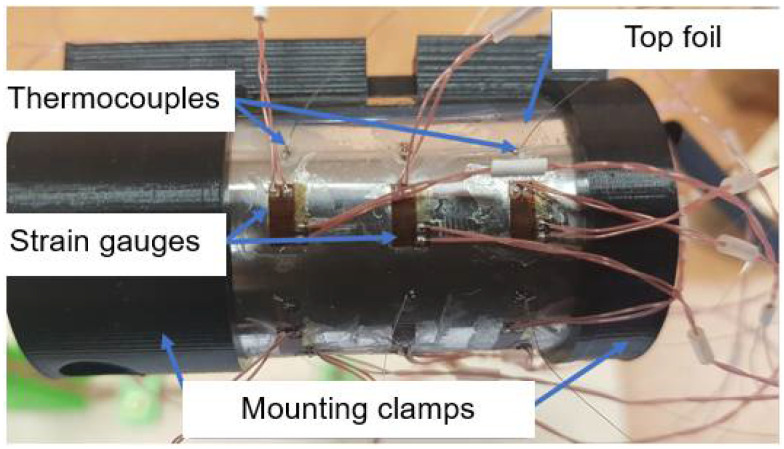
Specialized sensing top foil–the prototype fixed by the 3D printed mounting clamps prior to final assembly in the bushing. The strain gauges were not used during the present study.

**Figure 6 sensors-22-05718-f006:**
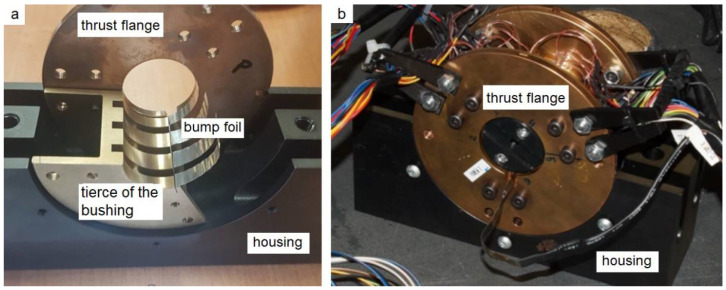
GFB prototype assembly: (**a**) bushing partially mounted in the housing with a single tierce and bump foil visible, (**b**) complete bushing with the sensor cabling.

**Figure 7 sensors-22-05718-f007:**
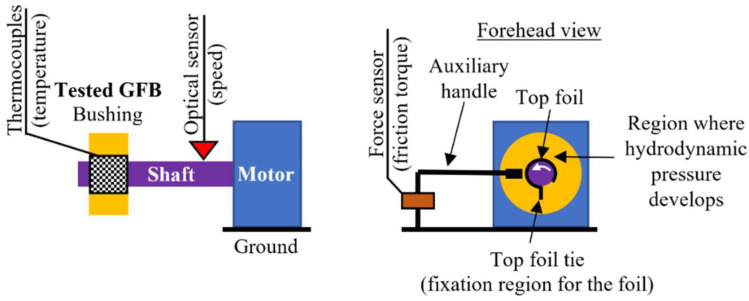
Schematic view of the freely-suspended configuration of a GFB.

**Figure 8 sensors-22-05718-f008:**
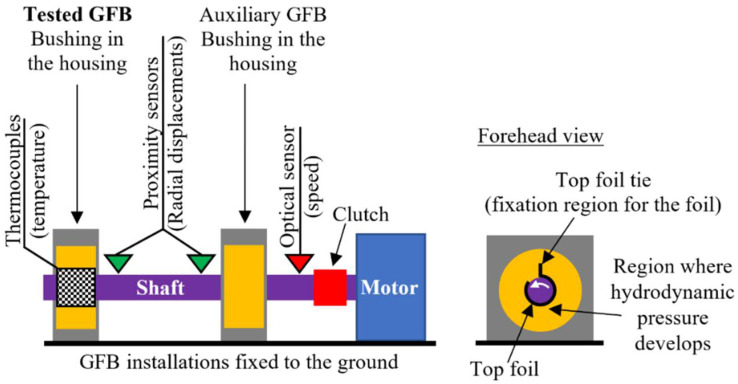
Schematic view of the two-node support configuration of a GFB.

**Figure 9 sensors-22-05718-f009:**
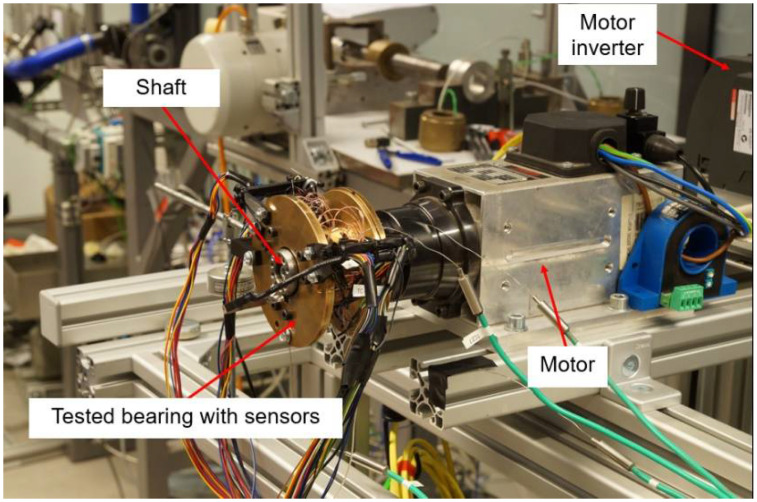
Test stand with the bearing inspected for the freely-suspended configuration.

**Figure 10 sensors-22-05718-f010:**
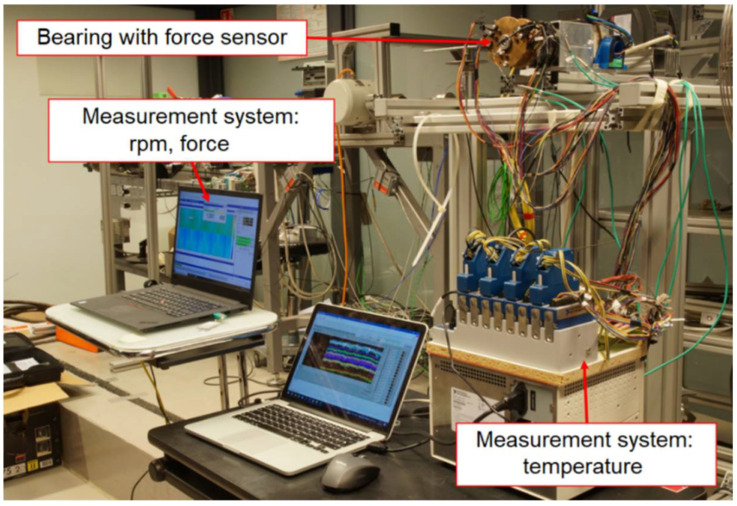
Measurement systems used for the freely-suspended configuration.

**Figure 11 sensors-22-05718-f011:**
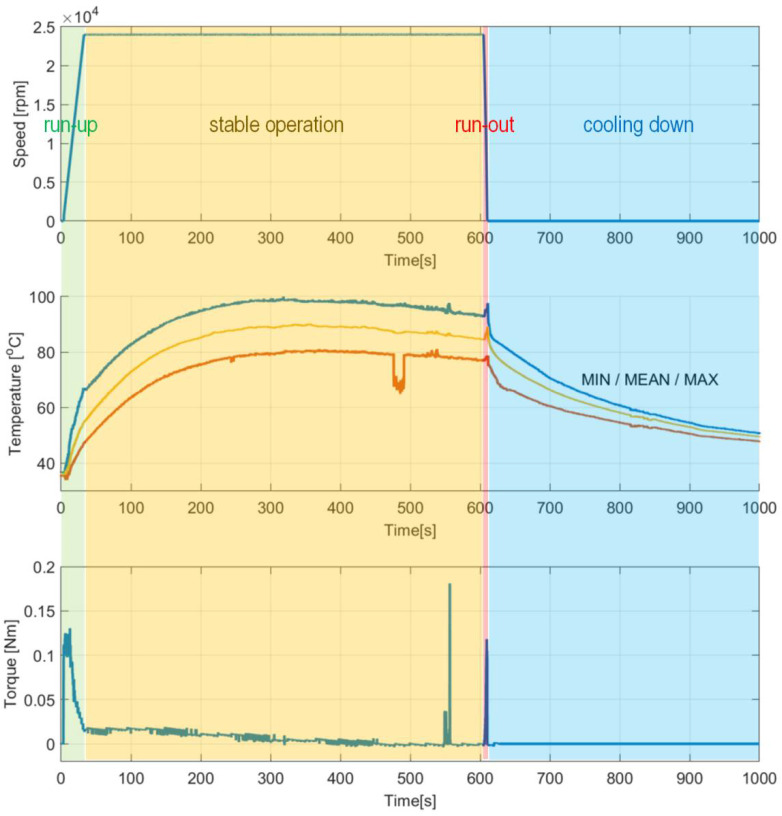
Temporal courses for an initial run conducted for the newly-installed top foil mounted in the tested GFB: (**top**) rotational speed, (**center**) temperatures and (**bottom**) friction torque. As regard with the temperatures’ courses, their minimum, mean and maximum values are respectively marked with three different colors. It should be noted that within the approximate time interval 470–490 s, the course indicating the minimum temperature of the bearing measured by all thermocouples suddenly drops. This results from the temporal loss of correct signal from one of the sensors installed. These outcomes are presented despite the measurement outlier occurrence since an initial run for a given top foil could be performed only once, i.e., no additional experiment would enable repeating the initial running-in procedure for a given set of the bearing’s foils.

**Figure 12 sensors-22-05718-f012:**
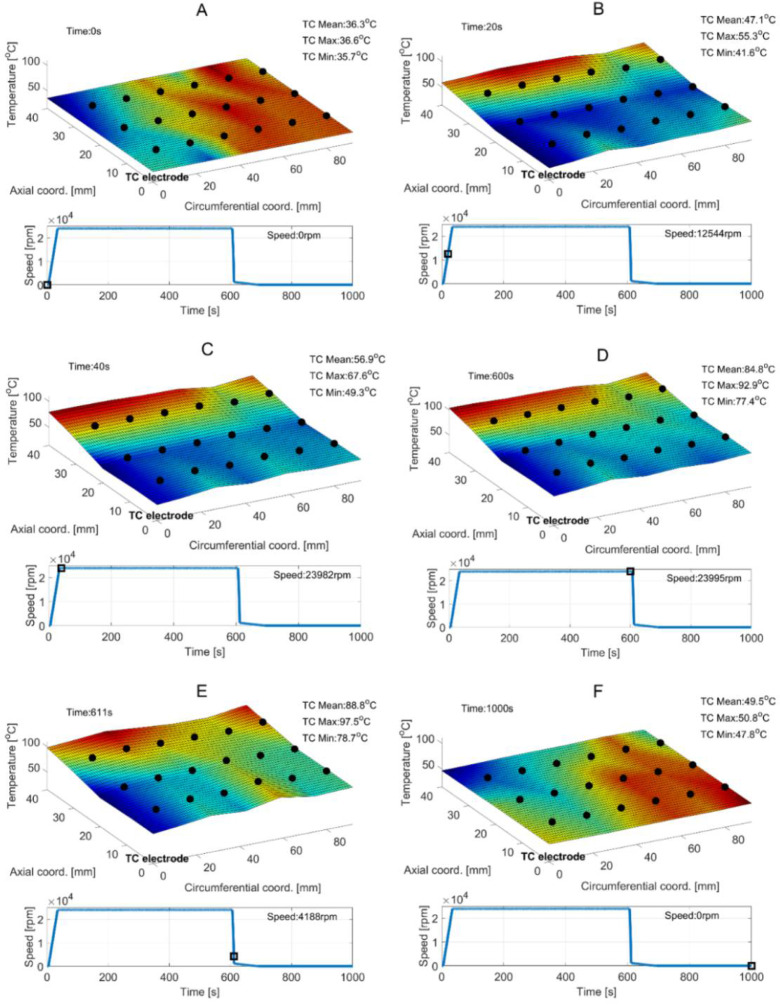
Experimentally identified temperature fields for the top foil found at given time moments–denoted as the subsequent cases (**A**–**F**). Black dots on the generated surface mark experimental readings for the thermocouples. Black squares located on the temporal courses reference the actual rotational speeds.

**Figure 13 sensors-22-05718-f013:**
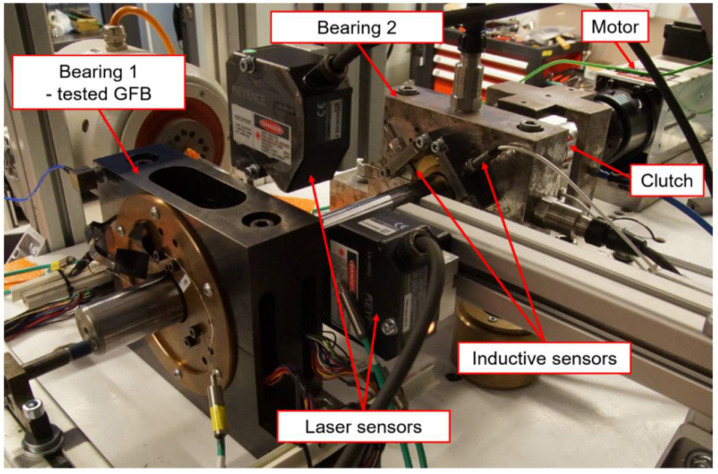
Test stand with the bearing investigated for the two-node support configuration.

**Figure 14 sensors-22-05718-f014:**
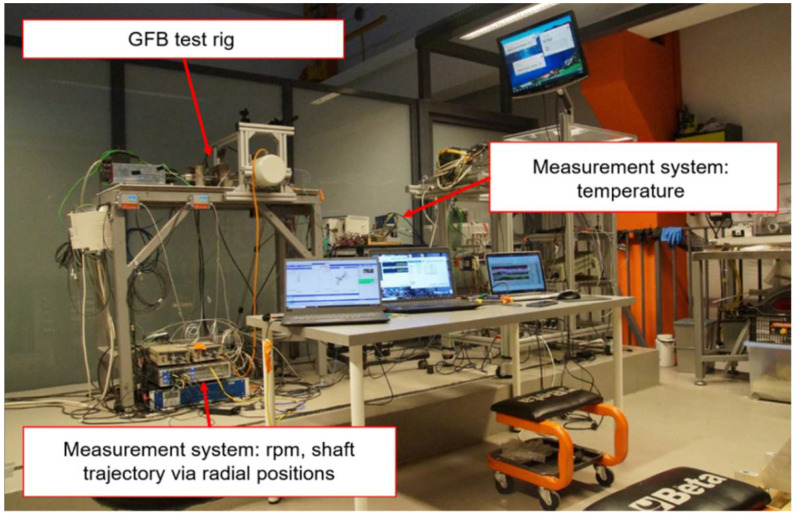
Measurement systems used for the two-node support configuration.

**Figure 15 sensors-22-05718-f015:**
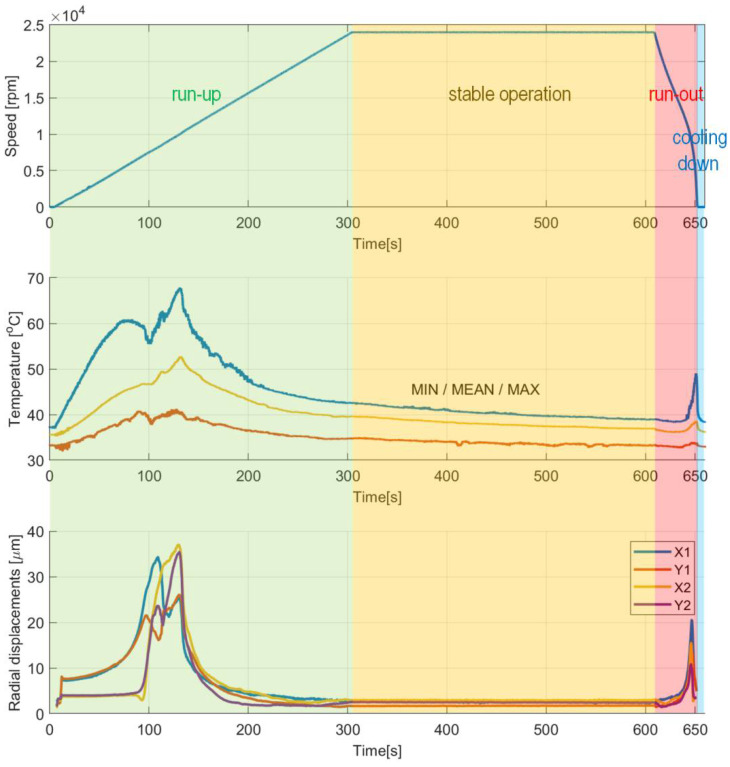
Temporal courses for: (**top**) rotational speed, (**center**) temperatures and (**bottom**) shaft’s radial displacements. Non-monotonic changes of the temperatures and radial displacements observed within the time interval 70–130 s indicate a gradually progressive process of adjustment for the set of foils and the accompanying air film development before it is entirely and continuously formed. The authors suggest that the identified temporal temperature drops follow the variation of the generated heat energy, originating from the evolving friction conditions. Next, however, still during the phase of a positive angular acceleration, which is present within the time interval 130–300 s, the values of the above-mentioned parameters considerably decrease, advantageously.

**Figure 16 sensors-22-05718-f016:**
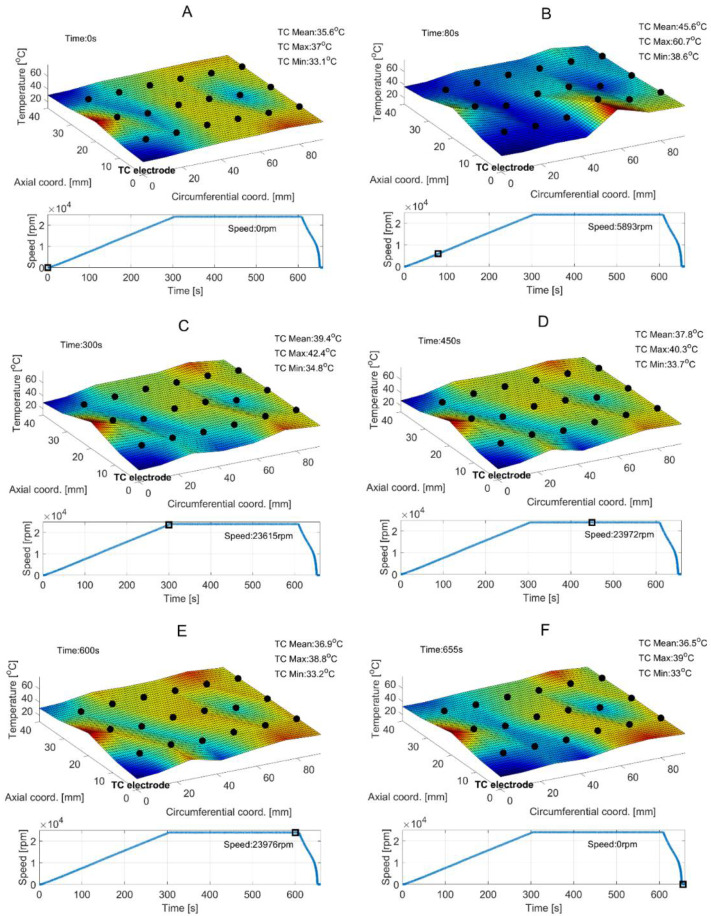
Experimentally identified temperature fields for the top foil found at given time moments–denoted as the subsequent cases (**A**–**F**). Black dots on the generated surface mark experimental readings for the thermocouples. Black squares reference the actual rotational speeds.

**Figure 17 sensors-22-05718-f017:**
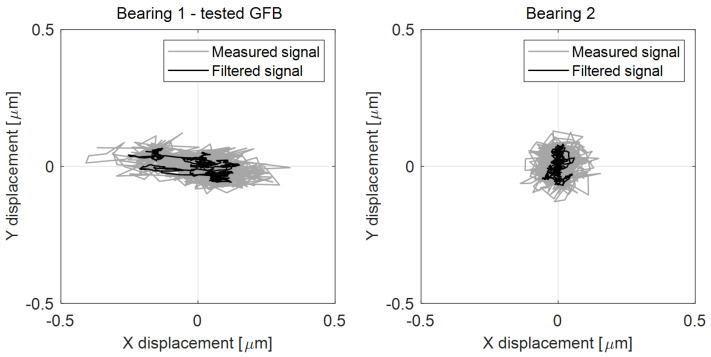
Shaft’s trajectories and their filtered counterparts registered for the stage of stable operation at 24,000 r/min (24 krpm). A moving-average filter was applied for smoothing the registered noisy data with the window size set to 10. The offset components were removed from the sensor readings so that the mean positions converge to the origin of the coordinate system.

**Figure 18 sensors-22-05718-f018:**
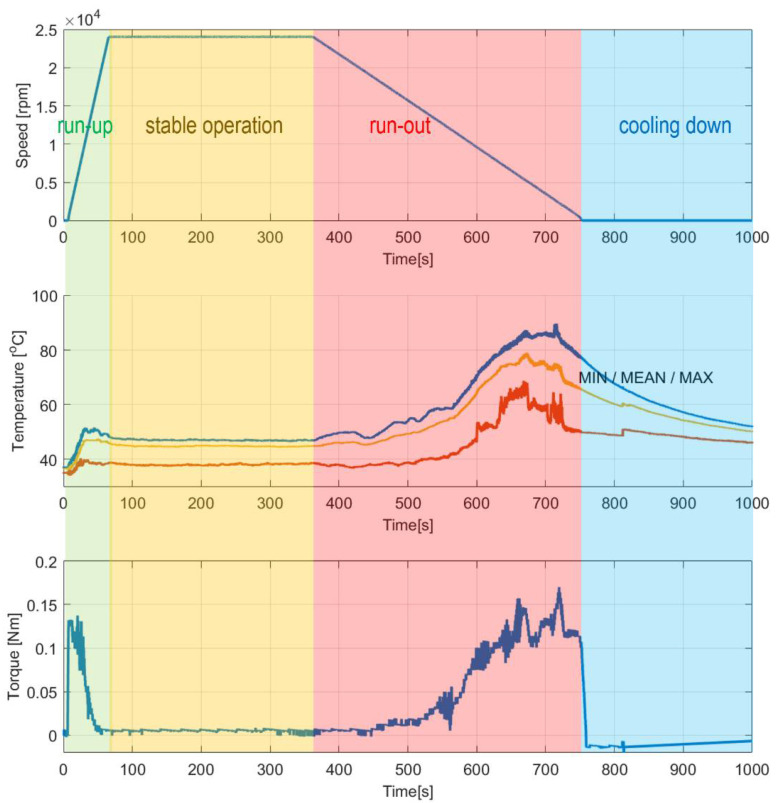
The effect of running-in the bearing’s top foil. In contrast to the temperature courses presented in Figure 11, constant readings for the mentioned quantity are obtained within the stable state stage of GFB’s operation. Moreover, as expected, the elevated temperature and friction torque are identified during both run-up and run-out stages, when developing and losing the air film, respectively. It is also worth noticing that a characteristic thermal stress release is observed for the bearing undergoing cooling down. Specifically, a progressive convergence to zero for the measured negative friction torque is observed due to the gradual reduction of the significant foils’ thermal deformations caused by the excessive heat generation during the intentionally extended period of the run-out stage of operation.

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
