# Peer review of "Thermal Characterization of a Gas Foil Bearing—A Novel Method of Experimental Identification of the Temperature Field Based on Integrated Thermocouples Measurements"

_sensors, 2022, doi:10.3390/s22155718_

Round 1

Reviewer 1 Report

This paper presents a method of the temperature identification based on the integrated thermocouples readings to determine the thermal properties of the specialized sensing top foil mounted in the tested bearing. The developed test stands successfully underwent the two measurement campaigns making use of both the freely-suspended and two-node support configurations. It is an interesting paper. However, the authors need to address the following minor issues before this manuscript is accepted:

1. In Introduction, the effect of the rotating shaft is discussed. As given in ‘Dynamic modeling and simulation of a flexible-rotor ball bearing system’, the deformations of the rotating shaft will greatly affect the internal contact characteristics of the support bearings. The above materials and references can be discussed too.

2. Some more descriptions for differences between the proposed method and listed references can be discussed, which can be used for showing the authors’ new contributions.

3. Some discussions for the test errors can be given in Section 6.

Author Response

Dear Reviewer,

Could you please see the attachment with our responses.

Kind regards,
Adam Martowicz
/on behalf of all authors/

Reviewer 2 Report

The paper presents an interesting method to measure the temperature distribution in GFBs.

However, it looks as if the sensors are placed at locations on the top foil under which there is no bump foil. As a result, the real bearing behavior is not represented at these locations either. It can be assumed that at these locations, due to the missing bumpfoil, the aerodynamic pressure in the gap causes the topfoil to bend slightly. As a result, the film height in these regions and thus the shear losses in the gap are lower than in the regions of the bearing with bump foil. This should also result in a lower temperature at the measurement location. Due to the interpolation of the temperature, it may well be that the existing temperature is thereby underestimated to a not irrelevant measure.

In the first experiment with a cantilevered bearing, the temperature rises very sharply. The authors explain this with the run-in process of the bearing. This could have been easily proven by repeating the same experiment with the bearing already run in. However, since this was not done, the increased heat development can also simply result from tilting of the bearing. This alternative explanation is also supported by the one-sided heat input in the axial direction. The picture of the test rig also casts doubt on the bearing's ability to self-align there.

The second test with a double-bearing rotor shows a significantly different temperature behavior. To explain this with the run-in process alone without showing experimental results to support it is rather questionable.

Apart from that, the authors have presented a good paper. One can follow the work relatively easily and the authors have also thought about many possible influences.

Author Response

(The authors gave the same response as above.)

Reviewer 3 Report

In the paper, the authors mainly described the setup of an air foil bearing with thermocouples installed in the top foil.

Even if the experimental setup is well described, there is a lack of novelty in the paper. Simple maps of temperature distribution are shown for run-ups with two different configurations.

It is hard to extract useful information from such result. Do the temperature distributions can be used for improving the bearing behavior or allow to highlight any hidden behaviors?

Does the temperature field agree with the numerical results?

Furthermore, there is not a comparison with other methods that can prove the advantage of using such bearing with temperature sensors.

Author Response

(The authors gave the same response as above.)

Reviewer 4 Report

I found your article very interesting, but in my opinion below remarks would improve your manuscript under the scientific level.

Comments and Suggestions for Authors:

1.       At the beginning, in the manuscript I would suggest to change the style of citing a few references in a row, exemplary [14,15,16,17] to [14-17].

2.       Regarding the description considering Figure 9 and Figure 10, I suggest to put it before the Figures.

3.       I found the incorrect notation of units in the manuscript, not in form of fraction.

4.       What is the reason of observed peaks in Figure 11 regarding the torque time-series?

5.       I suggest to introduce the table regarding the information on the applied software for data-acquisition, laser system and so on.

6.       In Figure 11 and 15 I suggest to mark the specific phase of dynamical behaviour for GFB, i.e. run-up, stable solution, on one of plots, exemplary for velocity or temperature. You mention about it in Section 6.

7.       Figure 17 – regarding obtained orbit plots I suggest to tighten it a little bit with axes and every chosen point to show the trajectory in a better manner.

8.       In the Introduction I suggest to cite 3 following references, that are discussing rolling-element bearings and one sentence more about the difference between rolling-element bearings and gas-foil bearings.

·       Bortnowski et al. (2022) – Roller damage detection method based on the measurement of transverse vibrations of the conveyor belt – Eksploatacja i Niezawodnosc – Maintenance and Reliability 24(3), pp. 510-521. http://www.ein.org.pl/2022-03-12

·       Ambrożkiewicz et al. (2022) – The influence of the radial internal clearance on the dynamic response of self-aligning ball bearings – Mechanical Systems and Signal Processing 171(8), 108954.

·       ZmarzÅ‚y (2020) – Multi-dimensional mathematical wear models of vibration generated by rolling ball bearings made of AISI 52100 bearing steel – Materials 13(23), 5440.

Author Response

(The authors gave the same response as above.)

Round 2

Reviewer 2 Report

My questions from the first review were answered in a reasonable way. Even more measurements have been added to clearly show the GFB run-in process. The paper is ready for publication in its revised state.

Reviewer 3 Report

The authors imroved their paper and gave satisfactory answers about the novelty of the paper.